# DANTZIG-WOLFE DECOMPOSITION AND DEEP REINFORCEMENT LEARNING

## ABSTRACT

The 3D bin packing problem is an NP-hard optimization problem. RL solutions found in the literature tackle simplified versions of the full problem due to its large action space and long episode lengths. We use a Danzig-Wolfe formulation to decompose the problem into a set partition and 3D knapsack problem. The RL agent is used to solve the 3D knapsack problem and CPLEX (a mixed integer linear programming solver) is used to solve the set partition problem. This removes the bin selection action from the action space of the agent and reduces the episode length to be the number of items required to fill 1 bin rather than all items in the inference. We thereby simplify the learning problem compared to the full 3D bin packing case. The trained agent is used at inference time to iteratively generate columns of the Danzig-Wolfe formulation using the column generation procedure. Improved solutions were obtained on 28/47 instances compared to an existing RL solution.

Tight lower bounds are required to guarantee that the feasible solutions provided by heuristics or RL agents (an upper bound) have achieved optimality (guaranteed if the upper and lower bounds are equal). We improved on SOTA lower bounds on 17/47 instanced by using exact solvers in both the master and sub-problem of the CG procedure. The lower bounds allow us to either guarantee optimality or estimate the optimality gap of the proposed solution combining an RL agent in the sub-problem with an exact solver in the master problem.

## 1 INTRODUCTION

Multi-dimensional variants of the bin packing problem, with added industrial constraints, are particularly challenging (Bortfeldt & Wäscher, 2013). In this work we consider stability constraints for cuboid items and containers. We solve the *offline 3D bin packing problem*. We define this to be a problem where all items are known in advance and must be packed in a minimal number of containers. This is in contrast to the *offline 3D knapsack problem* where the objective is to maximize volume utilization in a single container, with no constraint to pack all items. This distinction has not always been clear in the RL literature which often tackles the 3D knapsack problem (Bonnet et al., 2023) or versions of it that have no constraints on some or all of the container dimensions (Laterre et al., 2018; Zhang et al., 2021). These are also simplified versions of the problem since one does not need to close a container and open a new one to pack all items. Fang et al. (2023a;b) reduce the action space size by using RL to choose the item and a heuristic to choose the placement in a 2D problem. Other variants in the RL literature treat the *online 3D bin packing problem* where items are not known in advance, thereby reducing the action space of the RL problem (Zhao et al., 2021; 2023).

Iteratively solving a 3D knapsack problem (optimizing for volume utilization in 1 fixed size bin, then opening a new bin and doing the same with the remaining items) can provide a feasible solution but it is likely to be sub-optimal. This can be illustrated through a 1D example. Suppose we have 6 items of lengths 1 to 6 respectively, and bins of size 7. The optimal bin packing solution to this problem packs all items into 3 bins, each with 100% volume utilization. If we try to solve this by first solving the knapsack problem, packing items of length 4, 2 and 1 would maximize volume utilization in the full bin. However, this would force the remaining items to occupy 1 bin each, thereby resulting in using 4 bins instead of the optimal 3. This idea can clearly be generalized to the multi-dimensional

problem. The work of this paper allows us to solve the offline 3D bin packing problem using a combination of a knapsack RL solver and an exact solver without suffering from these sequentiality issues. The added value of our work can only be seen on instances where multiple containers need to be packed. The results will be equal to the RL agent under consideration for an instance that can fit into a single container.

Exact optimization methods have been developed for both the offline 3D bin packing and knapsack problems, but tend to scale badly or omit certain constraints. Martello et al. (1998) and Hifi et al. (2010) both proposed exact formulations for the full 3D bin packing problem but did not allow items to be freely rotated nor provide stability constraints. Junqueira et al. (2012) and Nascimento et al. (2021) proposed exact formulations for the offline 3D knapsack problem. Silva et al. (2019) compared many different existing methods on the 3D knapsack problem. Tsai et al. (2015) consider a single container problem where the volume of packed items is minimized, with no constraints on container dimensions. Paquay et al. (2016) consider packing multiple bins of non-cuboid shape with many constraints but the number of items in an instance does not exceed 27. Kurpel et al. (2020) extended Junqueira et al. (2012)'s formulation to tackle problems with multiple bins, though the compute time on large problems was long.

In the field of exact integer and mixed integer linear programming, decomposition methods are used to reduce a large problem into a collection of smaller ones that can be solved iteratively to solve the original problem. One example is Danzig-Wolfe decomposition into a master and sub-problem, solved using column generation (CG). Using a 3D knapsack problem as the sub-problem of a CG procedure can mitigate the sequentiality issue mentioned previously. However, CG relies on being able to rapidly solve the sub-problem, which is not possible for exact solvers on large 3D knapsack problems. Some researchers have replaced an exact solver for the sub-problem with heuristics or past data (Mahvash et al., 2017; Eley, 2003; Elhedhli et al., 2019; Duan et al., 2022; Zhu et al., 2012). In doing this, one can find good solutions fast, at the cost of losing the guarantees on the optimum and invalidating the lower bound achieved by the restricted master's linear relaxation at convergence. In this paper, we use an RL agent in the sub-problem instead of a heuristic. In a separate experiment, we used an exact solver in the sub-problem so that we could produce tight valid lower bounds in order to evaluate the RL+CG algorithm. The restricted master problem is a set partition problem and the sub-problem a 3D knapsack problem. We use an exact solver (CPLEX) for the master problem since this is not the bottleneck of the algorithm. The test cases used to evaluate the algorithm require multiple containers to pack all items. This is a challenging data set not yet tested by existing RL algorithms.

The first novel contribution of this paper is the extension of the Jumanji *BinPack* environment to accommodate the ability to place items in any of 6 orientations (rather than 1) and enforcing the constraint of supporting 100% of the area of the item's lower face by the floor or previously placed items. We also add a 'value' feature to items and the agent packs to maximize packed value instead of packed volume such as is currently achieved in the literature. The main novel contribution of the paper is proposing a new CG algorithm that combines RL and exact solvers for the full 3D bin packing problem. We demonstrate improved inference solutions compared to sequentially applying the 3D knapsack RL agent on remaining items of the problem. To the best of our knowledge, this is the first time RL algorithms have been tested on such test cases in the literature. We were also able to assess the quality of solutions with improved lower bounds which we obtained using exact solvers and CG. Finally, unlike existing RL solutions, we can solve instances that have multiple types of containers to choose from.

## 2 BACKGROUND

Pre-requisites for understanding the work of this paper are explained here. Section 2.1 explains the Dantzig-Wolfe formulation and CG algorithm in the context of the 3D offline bin packing problem. Section 2.2 references the advantages of the exact formulation of the 3D knapsack problem that we used from the literature within a CG framework to provide tight lower bounds. We then describe the open-source Jumanji environment in section 2.3 that was modified in our work (see section 3.1) to be used as part of the efficient RL+CG algorithm.

## 2.1 DANZIG-WOLFE REFORMULATION AND CG ALGORITHM

The Danzig-Wolfe formulation of the bin packing problem assumes that the integer linear program (ILP) to be solved is simply picking the best combination of packing configurations that satisfy the demand of each item. This is called the master problem. Packing configurations are assumed to be provided in the set $Y$ and respect all feasibility and industrial constraints. Since all the packing constraints are already taken into account by the set $Y$, the only integer variables in the master problem are those corresponding to the number of each packing configuration to be used in the final solution. The linear relaxation of this problem is therefore tight and gives a good lower bound to the full problem (Vanderbeck, 2000). Mathematically, the Dantzig-Wolfe formulation is expressed as follows:

$$f = \min_x c^T x \tag{1}$$

$$\text{s.t.} Ax = b \tag{2}$$

$$x \in \mathbb{Z}^n_{\geq 0} \tag{3}$$

$$A_j \in Y \quad \forall j \tag{4}$$

$$Y = \{A_j \in \mathbb{Z}^m_{\geq 0} : A_j = g(y), Dy \leq d, y \in \{0,1\}^k\} \tag{5}$$

$b$ is a 1D vector of size $m$, where $m$ corresponds to the number of item types in the instance to be packed. Each element $b_i$ of this vector corresponds to the demand of item type $i$. $A$ is a 2D matrix of size $m \times n$:

$$A = \begin{pmatrix} \uparrow & \uparrow & \uparrow & \uparrow & \cdots & \uparrow & \cdots & \uparrow \\ A_1 & A_2 & A_3 & A_4 & \cdots & A_j & \cdots & A_n \\ \downarrow & \downarrow & \downarrow & \downarrow & \cdots & \downarrow & \cdots & \downarrow \end{pmatrix}$$

Each column $(A_j)$ represents 1 packing configuration in a single bin, each element of the column provides the number of items of a given type in that packing configuration. For example, element $a_{ij}$ represents the number of items of type $i$ packed in container packing configuration $j$. $x$ is a 1D vector of size $n$ and each element $x_j$ represents the number of times the $j$th packing configuration is used in the solution. $c$ is a 1D cost vector of size $n$, element $c_j$ corresponding to the cost of the packing configuration $j$. In this case, the cost of a packing configuration is the cost of the container used in the solution, this is a constant for single container type problems. $Y$ represents the set of all possible container packing configurations that respect the packing constraints. $y$ represents a $k$ dimensional vector that represents all the variables required to define the 3D knapsack problem and its industrial constraints (defined by $Dy \leq d$) and the function $g(y)$ maps $y$ to the column vector that indicates the number of each item in the given packing configuration. $D$ and $d$ are the matrix and right-hand side vectors that represent the feasibility and industrial constraints of packing in a single bin.

Since $Y$ is extremely large, enumerating all columns of the full problem is as complex as the original problem, and the coupling constraint matrix $A$ becomes huge. The CG algorithm provides a way of enumerating only those members of $Y$ that are *useful* to the master problem, thereby reducing the size and complexity of the master problem significantly. The master problem with a subset of feasible solutions: $P \subset Y$ instead of $Y$ in equation 5 is called the restricted master problem (RMP). The column generation algorithm is illustrated on a trivial example in appendix F, the mathematical explanation will be explained here.

We can use a heuristic or RL agent to generate an initial set of feasible packing configurations, $P$, which are columns of the matrix $A$. Let's assume we start with an initial solution of $p$ packing configurations that satisfy the demand constraint 2. The RMP can be expressed as follows:

$$f = \min_x c^T_{[:p]} x_{[:p]} \tag{6}$$

$$\text{s.t.} A_{[:p]} x_{[:p]} = b \tag{7}$$

$$x_{[:p]} \in \mathbb{Z}^p_{\geq 0} \tag{8}$$

$$A_j \in P \tag{9}$$

$$P = \{\text{Packing configurations in initial solution}\} \tag{10}$$

We use the notation $[:p]$ to show that it is only elements (for vectors) or columns (for matrices) from 0 to $p$ that are included in the problem. The column generation algorithm adds a new column to the $A$ matrix at each iteration. So if there are 3 initial packing configurations, the traditional column generation algorithm will start with matrix $A_{[:3]}$ and have matrix $A_{[:3+l]}$ by iteration $l$.

$$A_{[:3]} = \begin{pmatrix} \uparrow & \uparrow & \uparrow \\ A_1 & A_2 & A_3 \\ \downarrow & \downarrow & \downarrow \end{pmatrix}, \quad A_{[:3+l]} = \begin{pmatrix} \uparrow & \uparrow & \uparrow & \uparrow & \cdots & \uparrow \\ A_1 & A_2 & A_3 & A_4 & \cdots & A_{3+l} \\ \downarrow & \downarrow & \downarrow & \downarrow & \cdots & \downarrow \end{pmatrix}$$

The LP relaxation of the RMP can be solved quickly by efficient algorithms such as Simplex (Dantzig & Thapa, 1997). This method also provides the dual values $\lambda$ (1D vector of size $m$) of the demand constraints (equation 7) which provide the marginal cost of increasing $b$. For example, $\lambda_i = \frac{\partial f}{\partial b_i}$ is the rate at which the cost would increase if the demand of $b$ increased. This gives us an idea of the "value" of an item in any potential packing configuration that we might add to the RMP. An intuitive explanation for this is that if $\lambda_i$ is positive, it means that there are not enough appropriately packed bins for item type $i$, so we would like the new candidate packed bin to have more of item type $i$. Similarly, if this value is negative, it means that the packed bins available in the RMP are good for items of type $i$ since we can increase demand and reduce the overall cost, therefore we do not need new candidate solutions with items of this type. A mathematical explanation of this can be found by looking at the Simplex method and seeing that adding a new variable (column) to the basis is beneficial to the objective function only if it has negative reduced cost $c_{p+l} - \lambda_{p+l-1}^T g(y_{p+l})$ (the subscript $l$ indicates the $l$th iteration of the CG).

The sub-problem then becomes a 3D knapsack problem that should propose a new packing configuration with minimal reduced cost in terms of the dual variables of the relaxed RMP, whilst still respecting all integer packing constraints. This would add a new candidate solution to $P$ in equation 10, thereby adding a new column to the demand constraint matrix and replacing $p$ in equations 6, 7, 8 by $p+1$. The best candidate solution is the one that solves the following optimization problem:

$$\min_{y_{p+1}} r = c_{p+1} - \lambda_p^T g(y_{p+1}) \tag{11}$$

$$\text{s.t.} D y_{p+1} \leq d \tag{12}$$

$$y_{p+1} \in \{0, 1\} \tag{13}$$

This is often referred to as either the sub-problem or the pricing problem. $r$ represents the reduced cost which is to be minimized over the packing variables $y$. Equation 12 represents the feasibility and industrial packing constraints as in the Danzig-Wolfe decomposition equation 5. $y_{p+1}$ represent the packing variables at this iteration of the algorithm. In the ILP definition of the knapsack problem, these are binary variables (see section 2.2).

The relaxed RMP is solved again with the added column, new dual variables are generated, which are in turn used in the sub-problem to generate a new column. This procedure continues until the optimal solution to the sub-problem is non-negative. This means that there are no new candidate solutions that can improve the objective function of the relaxed RMP. An optimal solution to the relaxed RMP is therefore found without having to generate all possible packing configurations. Note that this procedure obtains the optimal solution to the relaxed RMP and is therefore only a lower bound to the full problem with integer constraints. If this lower bound happens to be an integer solution then we know we have found the optimal solution. Otherwise, the RMP can be solved with the integer constraints and, provided that the number of columns remains reasonable, can be solved efficiently by an ILP solver such as CPLEX, Gurobi, or SCIP. Branching can then be done on the resulting integer solution, but this is out of scope for this work.

The Danzig-Wolfe decomposition algorithm works well for problems where a large number of columns are not required to obtain the optimal solution. For the bin packing problem, this will vary depending on the properties of the instance in question. It also works well when the decomposition results in sub-problems that can be solved efficiently to optimality. The 3D knapsack problem is also an NP-hard problem and can take a long time to solve, this means that the CG method can become intractable in practice. As mentioned in the literature review, other authors have replaced the exact sub-problem with alternative methods such as heuristics or historically packed examples. When these methods are used, we lose the optimality guarantee on the relaxed RMP and therefore its

interpretation as a lower bound for the integer RMP. We also risk prematurely stopping the iterative procedure by not solving to optimality. This is why we believe a well trained value-based RL agent will provide good candidate solutions quickly, thereby reducing the risk of early termination and leading to good quality final solutions.

## 2.2 EXACT FORMULATION OF THE KNAPSACK PROBLEM

The formulation implemented in this work is based on Kurpel et al. (2020)'s extension of Junqueira et al. (2012)'s formulation to allow items to be placed in any of 6 orientations. It is an integer linear programming (ILP) formulation that uses what they define as *normal patterns* (see appendix E) to discretize the space in a bin by defining all the possible sets of coordinates in which a box can be placed. This discretization is much more efficient than a uniform refined discretization of the space and does not remove possible feasible solutions from the solution space provided that a *full support* constraint is enforced. This constraint is a common one in industrial problems since it ensures vertical stability by only allowing items to be placed in such a way that the bottom area of the item is fully supported by items or the container floor below it. Many additional constraints are included in their formulation, but only those needed for the test cases of this paper were implemented for this work: the non-overlap constraints, support constraints and constraints that ensure no more than the available count of a given item is able to be packed into a given bin.

The normal patterns in this implementation result in efficient solutions for instances with few item types and is relatively robust to large counts (unlike the Hifi et al. (2010) formulation which duplicates identical items, thereby not making the most of this information). It is also more efficient for instances that have items that are not the same type but may have at least one of their dimensions being of the same length, or when items are relatively large compared to the size of the container.

## 2.3 RL FORMULATION OF THE SINGLE KNAPSACK PROBLEM

The Jumanji bin pack environment can only pack 1 bin and leaves items that do not fit into this bin unpacked in the final solution. The advantage of the 3D knapsack problem is that it is relatively easy to apply a dense reward function. For example, the dense reward option coded in Jumanji returns the volume of the placed item normalized by the volume of the container, this corresponds to the increase in volume utilization within the bin at that environment step. Such a dense reward would be difficult to implement for the full problem because we do not know the total volume of containers that will be packed at the end of the episode, making the choice of normalization difficult. The episode lengths of the 3D knapsack problem are also shorter than those of the full problem because an episode ends once one bin is full rather than after packing all items.

Item placement is achieved through the use of "Empty Maximal Spaces" (EMS). Items are always placed in the back bottom left corner of an EMS. An empty container has 1 single EMS and the first placed item has only one possible placement action. The environment then creates new EMSs after the placement of this item (3 in this case, one in the x, y and z directions). These EMSs can overlap, but if an EMS is completely consumed within another EMS over the course of the episode, it will be removed. This strategy for placement can reduce symmetries in the problem. The use of this EMS strategy ensures that the no overlap constraints are respected. However, the current Jumanji implementation does not enforce any support constraints. Edge cases such as a very large item being placed on top of a very small item is possible, and thereby the environment does not ensure vertical stability. Items of the same type are treated independently so the environment cannot take advantage of items of similar types having large counts for scaling purposes. The environment also does not allow free rotation of items, thus limiting the action space.

The current random generator included for the Jumanji environment generates a perfect instance (one whose optimal solution has a volume utilization of 100% with no remaining items). This is achieved by randomly cutting a container into smaller cuboids. This means that the issues resulting from a lack of freedom of orientation and support constraints are not often observed in practice on this distribution of instances. This is because the optimal solution does not require the rotation of items and is one where the support constraint is respected automatically. This isn't the case for the instances on which we test. We therefore need to add these constraints and extensions to the Jumanji environment as well as allowing for optimizing for value packed instead of volume utilization.

## 3 METHODOLOGY

The extensions made to the Jumanji *BinPack* environment to provide the Jumanji value based environment are detailed in section 3.1. How we implemented a general CG algorithm that can call either the value based RL agent or an exact solver for the sub-problem is described in section 3.2. Finally, in section 3.3 we describe the test cases and computational setup used for experiments.

### 3.1 EXTENSIONS TO THE JUMANJI *BinPack* ENVIRONMENT

Three main additions to the Jumanji *BinPack* environment were made: The cuboid item can be rotated into 6 valid orientations; A 100% support constraint is enforced on the environment; A value feature is added to all items and a dense reward corresponding to maximizing the value in the bin can be applied. We also wrote a new generator (see appendix D) to train on instances where the optimal solution has remaining items.

The first extension was applied by duplicating an item 6 times and using the original item mask to mask out rotations that would be invalid. All possible rotations of the item are masked out if that item has been placed. The second extension was made by changing the way EMSs are generated after the placement of an item. The EMS generated above the item is now limited to be the size of the placed item along the x and y axes and up to the top of the container in the z axis. Additional EMS merging logic was added in order to merge EMSs that end up being contiguous along the x and y axes after multiple item placements. The third extension was made by adding a feature to the item type. A dense reward feature was defined so that at each step, the reward is: $r_t = \frac{v_{placed\_item}}{\sum_{i \in \{i : v_i > 0\}} v_i}$, where $v_i$ represents the value of item $i$. The normalization of the item features within the network was done by dividing the item values by the maximum absolute value in the instance.

### 3.2 OVERALL MODULAR FRAMEWORK FOR CG AT INFERENCE TIME

---

**Algorithm 1:** CG algorithm for solving the 3D-BPP with real-world constraints

**Input:** containers, items ;
$patterns \leftarrow$ `generate_initial_patterns`(containers, items);
RMP $\leftarrow$ `set_partitioning_problem`($patterns$, items);
$converged \leftarrow$ False;
**while** NOT $converged$ **do**
  $prices, lower\_bound \leftarrow$ RMP.`solve`($patterns$, items, integer_constraints=False);
  $converged \leftarrow$ True;
  $best\_reduced\_cost \leftarrow \infty$;
  $best\_pattern \leftarrow Null$;
  **for** c $\in$ containers **do**
    subproblem $\leftarrow$ `3D_Knapsack_problem`(c, items);
    $new\_pattern, objective\_value \leftarrow$ subproblem.`solve`($prices$);
    $reduced\_cost \leftarrow cost($c$) - objective\_value$;
    **if** $reduced\_cost \leq best\_reduced\_cost$ **then**
      $best\_reduced\_cost \leftarrow reduced\_cost$;
      $best\_pattern \leftarrow new\_pattern$;
    **end if**
  **end for**
  **if** $best\_reduced\_cost \leq 0$ **then**
    $converged \leftarrow$ False;
    $patterns.add(best\_pattern)$;
  **end if**
**end while**
$feasible\_solution \leftarrow$ RMP.`solve`($patterns$, integer_constraints=True);
**Output:** $lower\_bound, feasible\_solution$;

---

The general algorithm is described in algorithm 1. The initial set of packing configurations (patterns) is obtained by calling the $generate\_initial\_patterns$ function on the instance in the first line of

the algorithm. In order to ensure that the algorithm is always able to provide a valid primal solution, this function should return any feasible solution to the full bin packing problem. We use the result of the greedy RL agent applied successively to remaining items for this purpose (which we refer to as **sequential greedy RL** from now on), though any heuristic could be used. The CG algorithm then improves on this solution by iteratively calling the exact solver for the relaxed RMP to obtain dual variables and either an RL agent or an exact solver for the sub-problem.

The implementation using the exact solver is used for the generation of lower bounds. Provided the exact solver can converge in reasonable time, the final iteration of the relaxed RMP provides a tighter lower bound than that obtained by dividing the sum of the volume of all items by the volume of the container. This is useful in section 4 for estimating how far the primal solution is from the optimum.

When an RL agent is used for the sub-problem, we tested 3 different variants:

**Greedy RL** consists of running the RL agent with a greedy policy where at each step the agent takes the action with the highest predicted probability. One pattern is added to the master problem per CG iteration, provided it has negative reduced cost.

**Stochastic RL** consists of running $N$ agents with a stochastic policy where instead of taking the action with the highest predicted probability, each agent samples an action following the probability distribution predicted by its policy at each step. All the patterns that are found by the agents are added as long as there is at least one pattern with a negative reduced cost.

**Finetuned RL** consists of retraining the pre-trained RL agent on the instance to solve for 5 training epochs[1] of training epochs and recorded the best packing pattern found during the training. After the fine-tuning is over, using the newly learned policy, apply the *stochastic RL* strategy on the instance and add all the patterns found by the stochastic agents and the best pattern found during the training.

### 3.3 TEST CASES AND COMPUTATIONAL SETUP

A data set proposed by Ivancic (1988) is used to evaluate the algorithms. It consists of 47 3D bin packing instances. The number of shapes ranges from 2 to 5 and the total item counts from 47 to 180. All test cases require multiple bins in their optimal solution. This is why the sequential RL algorithm can perform badly. When we refer to a **sequential RL** agent we refer to applying an RL agent (greedy or fine-tuned) on the problem as if it was a knapsack problem until no more items fit into the bin, the remaining items are then taken to form a new knapsack problem. This process is continued until all items are packed. As pointed out in the introduction, it is expected to lead to sub-optimal results and is the main motivation behind combining RL with CG. Any RL training, fine-tuning and the CG experiments with CPLEX in both the master and sub-problem were carried out on a computing cluster. Greedy sequential RL, CG with greedy RL and CG with 32 stochastic agents was carried out on a laptop. For details see appendix B.

## 4 RESULTS

We first present the training results for the value based RL agent (section 4.1). The new lower bounds obtained by using CG and CPLEX at both the master and sub-problem levels are then presented (section 4.2). Finally, the results of using the CG algorithm with CPLEX solving the master problem and various different RL agents for the sub-problems are presented (section 4.3).

### 4.1 TRAINING THE VALUE BASED RL AGENT

The A2C agent implemented in the open source Jumanji library for the *BinPack* environment was used (see full details in appendix C) with the extended environment, new reward function and generator described in section 3.1. Figure 1 shows an improvement in the value packed as the number of epochs increases. As expected, the greedy agent performs best for later epochs.

---

[1]The batch size of the training algorithm was reduced to 3, the number of environment steps before a policy update reduced to 15 and the number of learning steps per epoch reduced to 24 to reduce the computational cost of this step. All other hyper-parameters were kept the same as in the Jumanji github defaults.

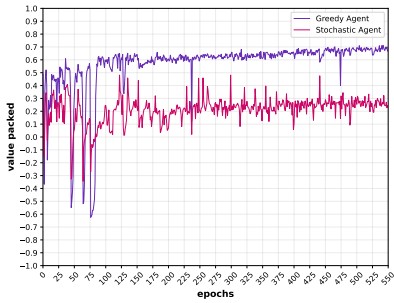

Figure 1: Average value packed over 40 evaluation environments after each training epoch.

## 4.2 New lower bounds on the solution

Table 3 shows the full results obtained by using the CPLEX solver for solving the exact formulations of the relaxed RMP and sub-problem at each iteration. The algorithm was warm-started by a solution obtained by packing all items in a set of containers by using the sequential greedy RL agent.

The result of the relaxed RMP at the final iteration gives a valid lower bound for the optimal solution. This lower bound is much tighter than the relaxed formulation of the full problem due to the smaller integer gap of the Danzig-Wolfe formulation, since only a subset of the integer variables are relaxed. We compare the lower bounds and primal solutions to those obtained by the best algorithms available in the literature. Note for different test cases, a different algorithm from the literature gives the best result, whereas we use the same algorithm for all instances.

We found an improved lower bound compared to SOTA lower bounds on 17/47 instances. This allowed us to confirm the optimality of SOTA solutions on 39 instances compared to 24 instances using SOTA lower bounds. SOTA equivalent primal solutions were found on 23 instances, and an improved solution on 2 instances. It is important to note that the aim of this section was to provide the best lower bounds for the solution which requires converging to the optimal sub-problem. This results in long compute times. Also, our implementation of the CG algorithm guarantees an optimal solution to the relaxed restricted master problem, branching would have to occur to further improve the resulting primal solution.

## 4.3 CG with a MILP solver for the master problem and RL agents for the sub-problems.

Table 4 shows the full results obtained using various RL agents to solve the sub-problem and the CPLEX solver for the RMP. These are compared to the sequential (see definition in section 3.3) greedy RL agent, the sequential fine-tuned RL agent and SOTA results currently found in the literature. Note that the SOTA result for each instance is obtained by different methods in the literature depending on which method gave the best solution for that particular test case, whereas we are comparing the same algorithm for all test cases.

Table 1: Aggregated results over the 47 test cases

|  | Sequential Greedy RL | Sequential finetuned RL | CG+ GreedyRL | CG+ 32StRL | CG+ Fine-TunedRL | Kurpel et al. (2020) | Zhu et al. (2012) |
|---|---|---|---|---|---|---|---|
| Total nb bins | 879 | 733 | 862 | 742 | 707 | 693 | 693 |
| Avg time (s) | 16.3 | 3191.7 | 31.1 | 178.0 | 704.9 | 880.9 | 116.7 |

The averaged results shown in table 1 show that gains compared to the sequential greedy RL agent can be made with very little extra time by using CG. In an average time of less than 12 minutes per run a total number of 172 bins can be saved over the 47 test cases. Even if results are required

quickly, 17 bins can be saved for an average extra time of less than 15 seconds. This shows that using CG is a way of breaking the disadvantages of sequentiality without increasing the action space and episode lengths of the RL agent.

The sequential fine-tuned RL highlights this even more. We fine-tune the packing on each bin sequentially. Although this algorithm should improve the packing in the initial bin, it will exacerbate the downsides of sequentiality. This is why it does not outperform the fine-tuned agent when used within the CG algorithm, and there are 4 examples where it performs worse (see table 4) than the sequential greedy RL. The timings for this agent are also very long.

Table 2: Summary of RL and CG Results

|  | Sequential Greedy RL | CG+ GreedyRL | CG+ 32StRL | CG+ Fine- TunedRL |
|---|---|---|---|---|
| Nb guaranteed optimal solutions | 10 | 10 | 15 | 25 |
| Nb improved solutions vs sequential greedy RL | - | 8 | 24 | 28 |
| Av. time of improved solution vs sequential greedy RL | - | 12.85 | 185.08 | 779.55 |
| Av. nb (max nb) of containers saved in improved solution vs sequential greedy RL | - | 2.13(3) | 5.71(55) | 6.14(55) |
| Nb of improved solutions vs SOTA | 0 | 0 | 0 | 1 |
| Nb of solutions at least as good as SOTA | 12 | 12 | 18 | 29 |
| Nb of solutions worse than SOTA | 35 | 35 | 29 | 17 |

Table 2 shows more specific information about the number of instances where an improved result was found. Averaged times and container savings are calculated only over those cases that have given an improvement. Large improvements are made on the RL results by combining RL with CG. Apart from in 1 case, these results do not currently beat SOTA (provided by either Kurpel et al. (2020) or Zhu et al. (2012) depending on the test case) for these test cases. Provided a realistic environment simulator is available, the RL agent can be more easily trained on much more complicated environments with more industrial constraints, where fine-tuned SOTA heuristics would become obsolete.

## 5 CONCLUSIONS

We conclude that combining RL with exact solvers via CG can improve on results found by existing RL solutions to the offline bin packing problem due to the sequentiality issue that they face. This was achieved without having to increase the action space of the RL agent nor complexify the reward function. The algorithm was tested on the OR Library data set and warm-started with a solution provided by the RL agent applied sequentially until all items are packed. CG+Greedy RL provides an improved solution in a reasonable time at inference on 17% of the instances, this was increased to 60% of instances when the RL agent was fine-tuned at each iteration of the algorithm. The fine-tuned RL only improved the obtained solution on 1 instance compared to SOTA results for these test cases. We expect the CG+RL method to outperform SOTA algorithms on very complex problems with more industrial constraints. The proposed algorithm can be used for test cases where multiple container types are available and one minimizes total container cost.

We obtained improved (tighter) lower bounds by using CG with an exact solver at both levels. A lower bound improvement was found on 36% of instances compared to the SOTA lower bound. This allowed us to confirm optimality for SOTA solutions on 32% more instances.

Further work could apply this method to test cases where a number of different types of containers are available for packing. The CG algorithm described would be particularly well adapted for this, since each iteration of the sub-problem would run a separate knapsack problem in parallel for each container type. Comparing this to the full RL problem with container choice in the action space would further demonstrate the advantages of using CG combined with a smaller RL problem in terms of training times and inference time solutions. Applying the method to more complex and constrained packing environments should also make it competitive with SOTA solutions. Applying Dantzig-Wolfe decomposition to other RL problems would also be an interesting avenue of research.

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

## A  APPENDIX: DETAILED RESULTS

Table 3: Results obtained from the comparison between the best-known solution and the solution obtained with the proposed algorithm on the evaluation dataset. Solutions correspond to the number of bins packed, optimal in bold.

| Id | Number of shapes | Number of items | SOTA lower bound | New lower bound | SOTA solution | Solution found by exact CG | Time taken (s) |
|----|------------------|-----------------|------------------|-----------------|---------------|----------------------------|----------------|
| 1 | 2 | 70 | 19 | 25 | **25**[a] | **25** | 17.8 |
| 2 | 2 | 70 | 7 | 9 | 10[a] | **9** | 32.5 |
| 3 | 4 | 180 | 19 | 19 | **19**[a] | **19** | 60.7 |
| 4 | 4 | 180 | 26 | 26 | **26**[a] | **26** | 31.2 |
| 5 | 4 | 180 | 46 | 51 | **51**[a] | **51** | 48.1 |
| 6 | 3 | 103 | 10 | 10 | **10**[a] | **10** | 23.7 |

[a]Solution found by Zhu et al. (2012) and Kurpel et al. (2020)

| 7 | 3 | 103 | 16 | 16 | **16**[a] | **16** | 24.2 |
|---|---|---|---|---|---|---|---|
| 8 | 3 | 103 | 4 | 4 | **4**[a] | **4** | 46.9 |
| 9 | 2 | 110 | 17 | 19 | **19**[a] | **19** | 30.6 |
| 10 | 2 | 110 | 43 | 55 | **55**[a] | **55** | 29.2 |
| 11 | 2 | 110 | 14 | 16 | **16**[b] | 17 | 37.2 |
| 12 | 3 | 95 | 53 | 53 | **53**[a] | **53** | 30.2 |
| 13 | 3 | 95 | 21 | 25 | **25**[a] | **25** | 31.0 |
| 14 | 3 | 95 | 27 | 27 | **27**[a] | 29 | 40.3 |
| 15 | 3 | 95 | 11 | 11 | **11**[a] | 12 | 45.4 |
| 16 | 3 | 95 | 21 | 26 | **26**[a] | 27 | 39.6 |
| 17 | 3 | 95 | 7 | 7 | **7**[a] | **7** | 3031.0 |
| 18 | 3 | 47 | 2 | 2 | **2**[a] | 3 | 15400.0 |
| 19 | 3 | 47 | 3 | 3 | **3**[a] | 4 | 22196.2 |
| 20 | 3 | 47 | 4 | 4 | **5**[a] | 8 | 476.8 |
| 21 | 5 | 95 | 17 | 20 | **20**[a] | **20** | 212.78 |
| 22 | 5 | 95 | 8 | 8 | **8**[a] | 11 | 18423.0 |
| 23 | 5 | 95 | 17 | 19 | 20[c] | 20 | 235.4 |
| 24 | 4 | 72 | 5 | 5 | **5**[a] | 7 | 24744.7 |
| 25 | 4 | 72 | 4 | 4 | **5**[a] | 6[d] | - |
| 26 | 4 | 72 | 3 | 3 | **3**[a] | 4[d] | - |
| 27 | 3 | 95 | 4 | 4 | **4**[a] | 5 | 22447.9 |
| 28 | 3 | 95 | 9 | 9 | **9**[b] | 11 | 377,4 |
| 29 | 4 | 118 | 15 | 16 | 17[a] | **16** | 601.1 |
| 30 | 4 | 118 | 18 | 22 | **22**[a] | **22** | 102.5 |
| 31 | 4 | 118 | 11 | 11 | 12[c] | 14 | 2690.1 |
| 32 | 3 | 90 | 4 | 4 | **4**[a] | 5 | 19641.1 |
| 33 | 3 | 90 | 4 | 4 | **4**[a] | 5 | 23869.6 |
| 34 | 3 | 90 | 7 | 7 | 8[a] | 10 | 212.5 |
| 35 | 2 | 84 | 2 | 2 | **2**[a] | 3 | 14959.4 |
| 36 | 2 | 84 | 10 | 14 | **14**[a] | **14** | 22.8 |
| 37 | 3 | 102 | 12 | 23 | **23**[a] | **23** | 25.8 |
| 38 | 3 | 102 | 26 | 45 | **45**[a] | **45** | 31.0 |
| 39 | 3 | 102 | 12 | 15 | **15**[a] | **15** | 921.9 |
| 40 | 4 | 85 | 7 | 7 | 8[a] | 9 | 118.3 |
| 41 | 4 | 85 | 14 | 15 | **15**[a] | 17 | 120.4 |
| 42 | 3 | 90 | 4 | 4 | **4**[a] | 5[d] | - |
| 43 | 3 | 90 | 3 | 3 | **3**[a] | 4[d] | - |
| 44 | 3 | 90 | 3 | 3 | **3**[c] | 4[d] | - |
| 45 | 4 | 99 | 2 | 2 | 3[a] | 3[d] | - |
| 46 | 4 | 99 | 2 | 2 | **2**[a] | 2[d] | - |
| 47 | 4 | 99 | 3 | 3 | **3**[a] | 4[d] | - |
| Sum | | | 596 | 682 | 691 | 723 | - |

Table 4: Comparison of Different RL Approaches to solve the 3D BPP. Solutions correspond to the number of bins packed, optimal in bold.

| Id | Sequential greedy RL | Sequential finetuned RL | CG with greedy RL | CG with 32 stochastic RL | CG with Fine-tuned RL |
|---|---|---|---|---|---|

[b]Solution found by Kurpel et al. (2020)

[c]Solution found by Zhu et al. (2012)

[d]The MILP could not be formulated due to lack of computational resources so the value of the solution recorded here is the value of the solution found after applying the RL agent sequentially on the instance

| | Sol | time (s) | Sol | Time (s) | Sol | time (s) | Sol | time (s) | Sol | time (s) |
|---|---|---|---|---|---|---|---|---|---|---|
| 1 | **25*** | 16.1 | 27 | 5694.7 | **25*** | 23.4 | **25*** | 41.7 | **25*** | 237.1 |
| 2 | 10 | 17.1 | **9*** | 1935.6 | 10 | 24.2 | 10 | 57.4 | **9*** | 429.4 |
| 3 | 23 | 27.7 | **19*** | 3862.8 | 23 | 35.1 | 20 | 122.7 | **19*** | 691.9 |
| 4 | **26*** | 27.0 | **26*** | 5322.0 | **26*** | 33.5 | **26*** | 112.3 | **26*** | 1011.4 |
| 5 | **51*** | 27.8 | 66 | 13788.8 | **51*** | 33.8 | **51*** | 65.3 | **51*** | 212.3 |
| 6 | **10*** | 17.8 | **10*** | 2178.9 | **10*** | 25.6 | **10*** | 78.9 | **10*** | 253.7 |
| 7 | **16*** | 18.2 | **16*** | 3375.7 | **16*** | 24.6 | **16*** | 98.7 | **16*** | 406.9 |
| 8 | **4*** | 17.8 | **4*** | 826.9 | **4*** | 25.4 | **4*** | 308.6 | **4*** | 654.8 |
| 9 | 25 | 18.8 | **19*** | 4035.1 | 25 | 27.5 | 22 | 69.8 | **19*** | 385.8 |
| 10 | 110 | 22.8 | **55*** | 11184.0 | 110 | 28.6 | **55*** | 53.2 | **55*** | 445.1 |
| 11 | 22 | 21.0 | **16*** | 3433.8 | 21 | 27.9 | 19 | 67.9 | **16*** | 1265.4 |
| 12 | 56 | 26.2 | **53*** | 9186.9 | 55 | 32.2 | **53*** | 106.8 | **53*** | 395.3 |
| 13 | 29 | 18.7 | **25*** | 5122.1 | 29 | 25.6 | 26 | 119.9 | **25*** | 647.7 |
| 14 | 33 | 20.2 | **27*** | 5600.0 | 30 | 32.3 | 28 | 117.0 | **27*** | 417.3 |
| 15 | 18 | 18.2 | **12*** | 2575.4 | 15 | 32.5 | **12*** | 678.5 | **12*** | 566.1 |
| 16 | 37 | 14.2 | 27 | 5557.0 | 37 | 20.7 | 33 | 80.4 | **26*** | 616.9 |
| 17 | 10 | 15.5 | **8*** | 1663.3 | 10 | 31.1 | **8*** | 263.91 | **8*** | 1116.1 |
| 18 | 3 | 9.5 | **2*** | 404.8 | 3 | 17.7 | 3 | 166.0 | 3 | 588.6 |
| 19 | 4 | 10.0 | **3*** | 639.0 | 4 | 31.4 | 4 | 245.4 | **3*** | 946.5 |
| 20 | 10 | 10.6 | **5*** | 1058.1 | 10 | 16.9 | **5*** | 134.9 | **5*** | 717.1 |
| 21 | 23 | 13.8 | 22 | 4642.7 | 22 | 25.2 | **20*** | 123.1 | **20*** | 765.8 |
| 22 | 11 | 15.3 | **9*** | 1865.6 | 11 | 22.1 | **9*** | 200.5 | **9*** | 1226.2 |
| 23 | 29 | 15.8 | **20*** | 4103.0 | 27 | 22.9 | 23 | 98.9 | **20*** | 1063.7 |
| 24 | 7 | 11.2 | **6*** | 1285.6 | 7 | 35.1 | **6*** | 222.4 | **6*** | 815.0 |
| 25 | 6 | 12.4 | **5*** | 1024.5 | 6 | 33.5 | 6 | 175.0 | **5*** | 1108.1 |
| 26 | 4 | 12.4 | **3*** | 654.3 | 4 | 68.0 | 4 | 452.6 | 4 | 932.4 |
| 27 | 6 | 15.9 | **5*** | 1091.0 | 6 | 22.3 | 6 | 234.1 | **5*** | 606.1 |
| 28 | 15 | 16.3 | **10*** | 2067.4 | 12 | 42.2 | 11 | 218.8 | **10*** | 653.6 |
| 29 | 29 | 16.0 | 18 | 3697.3 | 26 | 35.0 | 19 | 99.0 | **17*** | 731.5 |
| 30 | 28 | 15.3 | **22*** | 4356.6 | 28 | 21.9 | 25 | 68.8 | **22*** | 805.7 |
| 31 | 17 | 17.0 | **13*** | 2786.4 | 17 | 23.1 | 15 | 98.7 | **13*** | 825.6 |
| 32 | 5 | 14.5 | **4*** | 870.0 | 5 | 31.7 | **4*** | 337.1 | **4*** | 1282.9 |
| 33 | **5*** | 14.4 | **5*** | 1084.0 | **5*** | 29.1 | **5*** | 176.1 | **5*** | 874.9 |
| 34 | 10 | 14.7 | **9*** | 1915.8 | 10 | 26.9 | **9*** | 169.6 | **9*** | 620.9 |
| 35 | **3*** | 12.2 | **3*** | 647.7 | **3*** | 20.4 | **3*** | 206.3 | **3*** | 651.3 |
| 36 | **14*** | 12.3 | 16 | 3332.4 | **14*** | 19.6 | **14*** | 50.6 | **14*** | 225.5 |
| 37 | **23*** | 14.3 | 24 | 4997.4 | **23*** | 22.1 | **23*** | 91.9 | **23*** | 202.9 |
| 38 | **45*** | 16.9 | 49 | 9342.6 | **45*** | 22.8 | **45*** | 60.7 | **45*** | 198.5 |
| 39 | 16 | 16.0 | **15*** | 3204.3 | 16 | 22.6 | 16 | 72.7 | 16 | 201.7 |
| 40 | 13 | 14.2 | **9*** | 1814.5 | 13 | 27.3 | **9*** | 211.8 | **9*** | 714.3 |
| 41 | 26 | 17.1 | 16 | 3216.0 | 26 | 23.5 | 19 | 126.8 | **15*** | 1024.1 |
| 42 | 5 * | 12.7 | **5*** | 1066.5 | **5*** | 28.6 | **5*** | 244.0 | **5*** | 1006.4 |
| 43 | 4 | 13.5 | **3*** | 653.5 | 4 | 40.0 | **3*** | 405.1 | **3*** | 694.0 |
| 44 | **4*** | 14.3 | **4*** | 859.0 | **4*** | 41.1 | **4*** | 226.4 | **4*** | 1054.7 |
| 45 | **3*** | 15.3 | **3*** | 675.2 | **3*** | 55.3 | **3*** | 412.0 | **3*** | 1089.7 |
| 46 | **2*** | 13.4 | **2*** | 483.8 | **2*** | 83.4 | **2*** | 395.3 | **2*** | 1331.6 |
| 47 | **4*** | 15.0 | 4 | 831.1 | **4*** | 66.9 | **4*** | 292.2 | **4*** | 582.8 |

# B  APPENDIX: COMPUTATIONAL SETUP

For the training of the value based agent, a setup made of 8 Intel(R) Xeon(R) Platinum 8168 CPUs with a frequency of 2.70GHz, 32GB of RAM and 8 Tesla-V100-SXM3-32GB GPUs was used.

---

*Best solution found by an RL agent

For generating the optimal solutions using CG and a MILP solver for both the master and sub-problem, version 22.1.1.0 of the CPLEX commercial solver was used to solve the aforementioned problems. The setup on which the solver was run is made of 8 AMD-EPYC-7452 CPUs and 64GB of RAM. The total amount of time allowed for the algorithm to run on one instance was 18 000 seconds, whereas the maximum amount of time given for the solver of the sub-problem was 2 hours for each iteration of the algorithm. The solver of the master problem wasn't constrained in time since that problem isn't the bottleneck of this algorithm.

For generating the solutions using the CG with a MILP solver for the master problem and a greedy inference run of the trained value-based RL agent or 32 stochastic inference runs of the trained RL agents we used a laptop computer equipped with an 11th Gen Intel(R) Core(TM) i7-1165G7 CPU with 2.80GHz frequency, 16 GB of RAM and TigerLake-LP GT2 GPU.

The results of the CG algorithm with a MILP solver for the master problem and a finetuned RL agent were obtained by running the solver on a cluster equipped with 8 Intel(R) Xeon(R) Platinum 8168 CPUs with a frequency of 2.70GHz each, 64GB of RAM and one Tesla-V100-SXM3-32GB GPU.

## C  APPENDIX: TRAINING ALGORITHM AND NEURAL NETWORK ARCHITECTURE

The agent underwent training for 550 epochs using the same hyperparameters used by Bonnet et al. (2023) except for the batch size which was reduced from 64 to 8 (see table 5) because of the lack of computational power. The dense value-based reward function defined in section 3.1 was used. The policy of the actor agent was refreshed every 30 environment steps, this is referred to as a learning step. An epoch consists of 100 learning steps. Following each epoch, the actor agent's policy was assessed across 40 distinct evaluation environments. In both the training and evaluation environments, the value feature of each item is normalized by the maximum absolute value of all the items.

To train the deep RL agent the advantage actor-critic algorithm (A2C) was used. Table 5 gives the values of the hyperparameters passed to this algorithm.

| Hyperparameter | Value |
|---|---|
| Number of epochs | 550 |
| Number of learner steps per epoch | 100 |
| Number of environment steps per learner step | 30 |
| Total batch size | 8 |
| Learning rate | 1e-4 |
| Optimizer | ADAM |

Table 5: Hyper-parameter values used to train the RL agent

The architecture of the RL agent that was used to solve the sub-problem is the same architecture used by Bonnet et al. (2023) in their article introducing the Jumanji suite of environments. The agent receives as observation the set of empty maximal spaces (EMS) and the set of remaining items to pack in the container. To construct the embeddings for the items and the EMSs, we pass the observations of the agent through a transformer layer. The transformer is composed of two layers of stacked multi headed attention. Each attention layer is composed of an independent self-attention layer that is used for both of these sets, a cross-attention layer is used between each set based on whether an item fits in the corresponding EMS. Each one of these two multi-headed attention layers is composed of eight heads, their key size is 16 and the MLP layer is composed of 512 units. The resulting embeddings are used by both the actor and the critic. The final layer of the actor passes the embeddings through a linear layer and then applies an outer product on the resulting sets to ensure permutation equivariance, the output of this layer is the value of each action available to the agent. The actions that don't respect the constraints of support are masked to the agent by making their probability equal to 0.

## D    APPENDIX: DETAILS ON THE INSTANCE GENERATOR

In order to train our agent to solve instances of the 3D knapsack, The instance generator used by the BinPack environment of the Jumanji suite had to be modified to generate instances that don't entirely fit in one container. This kind of instance is necessary because the volume of the items in the instances of the sub-problem that the RL agent solves will inevitably be bigger than the volume of a container since ultimately we are trying to solve an offline 3D bin packing problem where the optimal packing can involve several bins. The generator takes as input an integer value representing the number of containers we want to cut to generate the items. A first set of items $S1$ is then generated by cutting the space inside one container. Another set $S2$ is created by duplicating the items of $S1$ until the target volume is attained. The values of the items are sampled randomly using a normal distribution with a mean of 0 and a standard variation of 1. This new generator ensures that our agent sees a diversified set of instances during its training.

## E    APPENDIX: NORMAL PATTERNS

Normal patterns are a discretization scheme that is used to discretize the three axes of the container that consider all the possible points of placing any item along one axis of the container. They were first used by Junqueira et al. (2012) to solve the problem of 3D knapsack with cargo stability and load-bearing constraints. Kurpel et al. (2020) then extended these patterns to include all possible rotations of items. A full mathematical explanation of these patterns can be found in either reference.

## F    APPENDIX: ILLUSTRATIVE EXAMPLE OF THE COLUMN GENERATION ALGORITHM

This section presents an example where the proposed algorithm is used to solve an instance of the offline 3D bin packing problem with stability constraints. Consider the instance of the problem where an infinite number of containers of size $10 \times 10 \times 10$ are available to pack the following items:

- 4 items of type 1 with dimensions $10 \times 10 \times 3$.
- 6 items of type 2 with dimensions $10 \times 10 \times 5$.
- 3 items of type 3 with dimensions $10 \times 10 \times 7$.

By kickstarting the algorithm with the packing patterns shown in figure 2, the value of matrix $A_{[:3]}$ in equation 7 is:

$$A_{[:3]} = \begin{pmatrix} 0 & 0 & 1 \\ 0 & 1 & 0 \\ 1 & 0 & 0 \end{pmatrix} \tag{14}$$

Each column of the matrix in equation 14 corresponds to a packing configuration/pattern (packing patterns $a$, $b$, $c$ in figure 2 can be seen in column order in the matrix). The rows correspond to the number of item types of that row index in each packing pattern: there is one item of type 3 in packing pattern $a$, one item of type 2 in packing pattern $b$ and one item of type 1 in packing pattern $c$.

The solution obtained to the set partitioning problem: the relaxed RMP (equations 6-10, with the integer constraint relaxed) at this iteration is 13 packed bins: 3 times the packing pattern $a$, 6 times the packing pattern $b$, and 4 times the packing pattern $c$. The dual costs associated with each one of the constraints of the set partitioning problem (equation 7) are equal to 1. Passing these items with a value equal to 1 to the knapsack problem will produce a new packing pattern where three items of type 1 are packed inside the container. This packing pattern is added to the set of available packing patterns (see figure 3) and the set partitioning problem is solved again with these packing patterns, represented by the new constraint matrix:

$$A_{[:4]} = \begin{pmatrix} 0 & 0 & 1 & 3 \\ 0 & 1 & 0 & 0 \\ 1 & 0 & 0 & 0 \end{pmatrix} \tag{15}$$

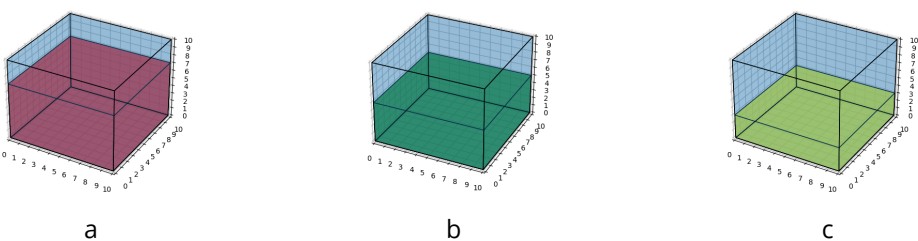

Figure 2: Packing patterns available before the first iteration of the algorithm

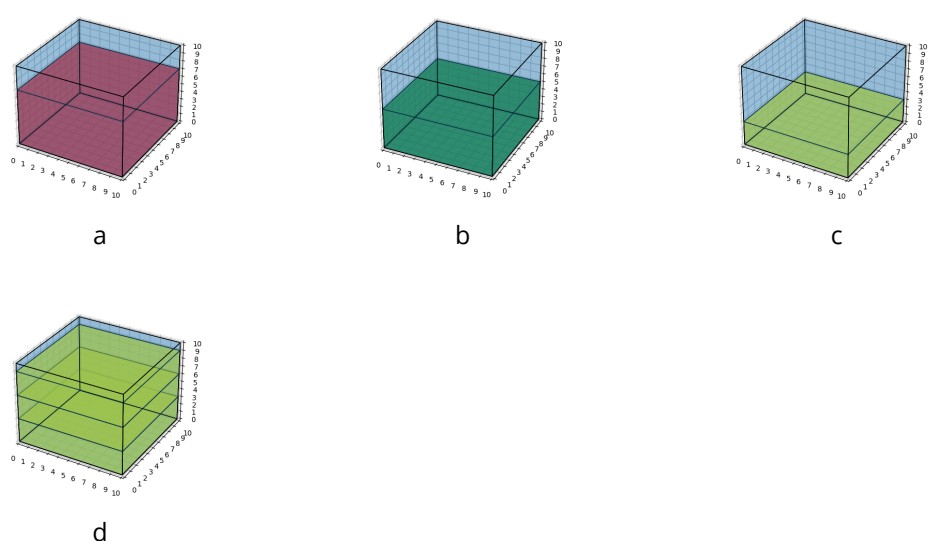

Figure 3: Packing patterns available after the first iteration of the algorithm

Solving the relaxed RMP at this iteration yields a solution that uses 3 times the packing pattern $a$, 6 times the packing pattern $b$, and 4/3 times the packing pattern $d$. Notice that packing pattern c is not used at all in the solution. The dual costs associated with each demand constraint are:

- 1 for the demand constraint on the items of type 1.
- 1 for the demand constraint on the items of type 2.
- 1/3 for the demand constraint on the items of type 3.

Solving a knapsack problem with these dual costs as item values generates a new packing pattern that is added to a set of available patterns. The new set of packing patterns is shown in figure 4. And adding the corresponding new column to the constraint matrix of the relaxed RMP gives:

$$A_{[:5]} = \begin{pmatrix} 0 & 0 & 1 & 3 & 0 \\ 0 & 1 & 0 & 0 & 2 \\ 1 & 0 & 0 & 0 & 0 \end{pmatrix} \tag{16}$$

Using these patterns, the solution to the relaxed RMP is a solution that uses 3 times the packing pattern $a$, 3 times the packing pattern $e$ and 4/3 of packing pattern $d$. The dual costs associated with the demand constraints of this solution are as follows:

- 1 for the demand constraint on the items of type 1.
- 1/2 for the demand constraint on the items of type 2.

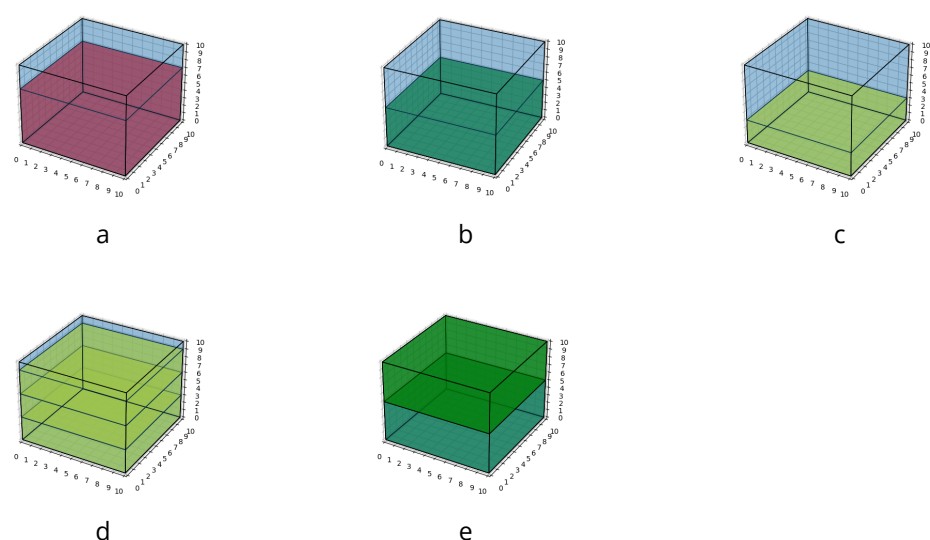

Figure 4: Packing patterns available after the second iteration of the algorithm

- 1/3 for the demand constraint on the items of type 3.

After solving the knapsack problem with these dual costs as item values, a new packing pattern is added to the available set to give the patterns shown in in figure 5. After adding this packing pattern

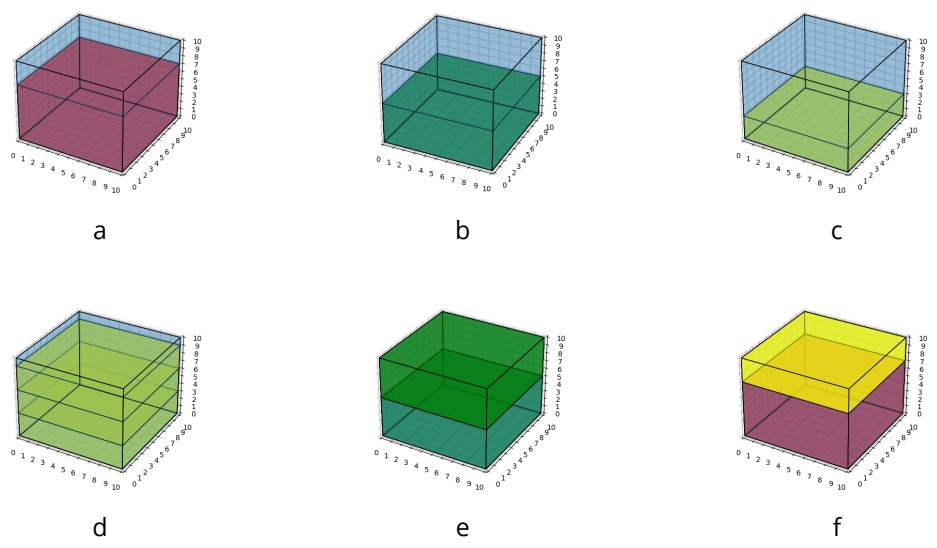

Figure 5: Packing patterns available after the third iteration of the algorithm

as a column in the constraint matrix we obtain:

$$A_{[:6]} = \begin{pmatrix} 0 & 0 & 1 & 3 & 0 & 1 \\ 0 & 1 & 0 & 0 & 2 & 0 \\ 1 & 0 & 0 & 0 & 0 & 1 \end{pmatrix}. \tag{17}$$

When the algorithm has access to all these packing patterns. The solver of the restricted master problem can find a solution using 7 containers by using 3 packing patterns of type $f$, 3 packing

patterns of type $e$, and 1 packing pattern of type $c$. The dual costs for each of the constraints after finding this solution are:

- 2/3 for the demand constraint on the items of type 1.
- 1/2 for the demand constraint on the items of type 2.
- 1/3 for the demand constraint on the items of type 3.

After solving the knapsack problem with these dual costs as item values the optimal solution doesn't exceed the cost of the container, thus we can conclude that the algorithm has converged after 3 iterations to a solution of value 7 packed bins. As this is an integer solution, we have an upper bound on the optimum solution. The solution to the RMP once the column generation procedure has converged is a lower bound on optimum solution. As the lower bound equals the upper bound, we know that 7 packed bins is the optimal solution to the problem.

