# OpenReview forum: "Dantzig-Wolfe Decomposition and Deep Reinforcement Learning"
_ICLR.cc/2024/Conference — Submitted to ICLR 2024_

### Official Review · Reviewer_TSyU · 2023-10-30

**Soundness:** 3 good
**Presentation:** 3 good
**Contribution:** 2 fair
**Rating:** 3
**Confidence:** 5

**Summary:**

The bin packing problem becomes even more complex when it involves multidimensional items and additional industrial constraints. While exact optimization methods exist for cuboid bin packing, they struggle with scalability and often neglect certain constraints, such as item rotation and stability. The paper proposes a column generation algorithm using the Danzig-Wolfe decomposition approach that incorporates both exact solvers and RL agents, offering improved lower bounds and more efficient solutions for the full 3D bin packing problem. The restricted master problem is solved by RL.

**Strengths:**

Multidimensional bin packing has not received enough attention and thus the choice of the topic is adequate. Capturing of complex side constraints is without doubt an important practical aspect.

**Weaknesses:**

The main weakness is the lack of contributions. For decades we know how to formulate standard bin packing by using DW and how to conduct column generation. It is also acceptable for a long time that the pricing problem can be solved as a relative simple IP or by RL (after all knapsack is sometimes solved as RL and the knapsack problem is the pricing problem in standard bin packing). I do not find anything innovative on the algorithm side in the paper. Apply RL in pricing in their specific context (multidimensional bin-packing) does not yield any novel contributions.

**Questions:**

Why did you use Cplex instead of Gurobi (which is now considered more efficient than Cplex - both have free licenses for academia)?
Column generation is known to have stability issues. There are known stabilization techniques. I wonder why they have not been tried.
Branch-and-price is a well established and in many cases the most effective algorithm for solving such problems (it combines branch-and-bound with column generation). Why is this methodology not being used as a benchmark?

---

> ### Author Response · Authors · 2023-11-17
> **Thank you for pointing out the strengths you found in our paper. Your feedback made us realize that we did not highlight the novel aspects of our paper sufficiently in the introduction and we hope that this is rectified in the new draft. We would like to address your questions and comments in the following reply.**
>
> Small correction in the summary of the reviewer, it is the sub-problem that is solved using RL not the master problem.
>
> # Comment on contributions:
> We made the literature review more concise in order to extend the final paragraph of the introduction to highlight all of the novel contributions of the paper in more detail:
>
> > The first novel contribution of this paper is the extension of the Jumanji BinPack environment to accommodate the ability to place items in any of 6 orientations (rather than 1) and enforcing the constraint of supporting 100% of the area of the item's lower face by the floor or previously placed items. We also add a ‘value' feature to items and the agent packs to maximize packed value instead of packed volume such as is currently achieved in the literature. The main novel contribution of the paper is proposing a new CG algorithm that combines RL and exact solvers for the full 3D bin packing problem. We demonstrate improved inference solutions compared to sequentially applying the 3D knapsack RL agent on remaining items of the problem. To the best of our knowledge this is the first time RL algorithms have been tested on such test cases in the literature.  We were also able to assess the quality of solutions with improved lower bounds which we obtained using exact solvers and CG. Finally, unlike existing RL solutions, we can solve instances that have multiple types of containers to choose from.
>
> It is true that the IP solution to the pricing problem is simple in 1D but no IP solution to the 3D problem has resulted in solutions within a reasonable amount of time on larger instances. This is why all existing CG solutions to the 3D bin packing problem have used heuristics in the sub-problem. Heuristics are notoriously efficient for specific applications of the problem but often become obsolete with the addition or omission of a constraint. RL solutions are more robust to this and can be retrained on more or less complex packing environments.
>
> Existing RL solutions have not been tested on test cases that involve packing more than one bin, since those in the literature are formulated as 3D knapsack problems with or without constraints on the dimensions of the bin. Our paper shows that these algorithms perform badly on multiple bin problems and can be improved using the CG algorithm without increasing the action space of the agent. The reason why was illustrated with a 1D example in the introduction for education purposes. We also extended the RL environment, the features seen by the agent, the instance generator for training and the reward used for the agent (maximise value packed rather than volume utilisation) so that it could be used as the pricing problem of the CG problem.
>
> # Answers to your questions:
>
> ## Why was CPLEX used instead of Gurobi:
> In our experience the differences in performance between CPLEX and Gurobi are problem dependent, both are state of the art solvers.
> The aim of the paper was to show how the use of an exact solver in the master problem, combined with RL in the sub-problem, could improve on the existing RL solution being used on its own. The master problem is easy to solve therefore the solver being used is not the bottleneck. In fact, we also tested with SCIP for the master problem with very little slow down. We didn’t include these results because they did not add to the conclusions of the paper.
>
> ## Comment on stability mitigation strategies for the CG algorithm:
> Thank you for this question. In our case, adding multiple columns in the stochastic RL+CG and fine-tuned RL+CG were in fact ways of improving the stability of the column generation algorithm.
>
> ## Comment on why branch and price was not used in this paper:
> Branch and price was not used because using RL in the sub-problem means that we can no longer guarantee that the final iteration of the column generation procedure is a valid lower bound, which is what the branching procedure relies on. The experiments using CPLEX in the subproblem did guarantee a lower bound and this was the purpose of those experiments, therefore there was also no need to branch. Solving the column generation without branching took a long time by itself, therefore we know this method would not be competitive with the benchmarked heuristics or any of the RL solutions in terms of compute time. We have modified the abstract to make the concept of bounds clearer in our context.

---

### Official Review · Reviewer_zq6L · 2023-10-31

**Soundness:** 2 fair
**Presentation:** 1 poor
**Contribution:** 2 fair
**Rating:** 3
**Confidence:** 4

**Summary:**

This article delves into the fusion of Danzig-Wolfe Reformulation with the Column Generation (CG) algorithm to address the 3D bin packing problem.

**Strengths:**

Strengths:

1. The paper introduces the integration of Reinforcement Learning (RL) with traditional mathematical programming methods to tackle the three-dimensional bin packing problem. This approach seems feasible.

2. By employing a combination of the Column Generation algorithm and RL algorithm to solve the packing problem, promising results were achieved. The use of the CG algorithm improved the lower bound of the original solution, offering a tighter lower bound.

**Weaknesses:**

Weaknesses:

1. The methods of Danzig-Wolfe Reformulation and Column Generation are existing approaches. The authors did not contribute significantly by improving these methods or specifically addressing the bin packing problem, rendering the contribution somewhat lacking.

2. Section 2.1, the core of the authors' method, is hard to comprehend. It is necessary to incorporate some illustrations and motivation to enhance readability.

3. The author's comparison with baseline methods is limited. They used a dataset introduced in 1988, which is quite outdated. They should consider recent methods like 'Attend2pack: Bin packing through deep reinforcement learning with attention' and new datasets (BED-BPP: Benchmarking dataset for robotic bin packing problems, Learning Efficient Online 3D Bin Packing on Packing Configuration Trees) to compare their proposed method in terms of time cost and performance.

**Questions:**

There is an issue of local optimal solutions when employing the RL algorithm to solve subproblems. Therefore, after the author decomposes the full problem into a set partition, how can they ensure that RL will inevitably find a solution?

---

> ### Author Response · Authors · 2023-11-17
> **Thank you for pointing out the strengths of our paper and for constructive feedback. In the following reply we respond to your questions, show how we addressed the weaknesses of the initial draft and explain why we do not agree with one perceived weakness described in the review.**
>
> # Comments on reported weaknesses of the initial draft:
>
> ## Weakness 1:
> We have extended the final paragraph of the introduction to better highlight the novel contributions of the paper. We agree that Dantzig-Wolfe and Column Generation are existing approaches, this is the first time they have been combined with an RL agent in the sub-problem. Existing RL solutions to the offline bin packing problem have thus far only tackled simpler instances that only pack 1 bin with a reduced number of constraints. Such RL solutions struggle to handle offline bin-packing instances where more than 1 bin needs to be packed in the optimal solution. Our paper is the first example we know of in the literature that tests existing RL solutions on such instances. We show that RL without CG struggles to obtain good solutions, whereas the CG algorithm helps the RL obtain better results.
>
> ## Weakness 2:
> We have improved our explanation in 2.1 by adding more intuitive explanations to the maths as illustrated matrices as a visual aid. We reference appendix E where the algorithm is demonstrated on a trivial example for clarity.
>
> ## Weakness 3:
> We do not agree that our baselines and test cases are lacking. The suggested baselines are not as challenging for the RL agent as those we used. We hope to explain why in the following paragraphs.
>
> The test cases we used have not yet been used in the RL literature because of their complexity. These are offline bin packing test cases with the constraint of packing all items in containers. The optimal solutions for these test cases require packing multiple bins, something that has not been handled by previous RL offline bin-packing solutions as they solve the 3D knapsack problem (with or without constraints on container dimensions). We hope to have made this clearer in the revised introduction by explaining in more detail the difference between the offline 3D bin packing problem and the offline 3D knapsack problem. Also, in the final paragraph of the introduction we highlight the novelty of using these test cases with RL algorithms.
>
> Unfortunately the dataset found in “BED-BPP: Benchmarking dataset for robotic bin packing problems” is not relevant to this work as it involves filling only 1 bin. The results would therefore be equivalent to the result of solving 1 subproblem of the CG algorithm (3D RL knapsack problem). The RL+CG algorithm is required if more than one bin is needed to optimally pack the whole instance. This is why we chose the more complex test cases from the late 80s. This comment made us think that we needed to clarify even more the difference between the knapsack and full bin-packing problems. We hope that the updated first paragraph in the introduction helps with this.
>
> We have used a recent RL method implemented by the Jumanji suite of environments for their recent paper (Clément Bonnet et al. 2023) and available in their open source github. We were able to modify the environment to accommodate the added constraints required by the test cases. Using Attend2Pack would have also required modification since the test cases we use have fixed container dimensions and the  implementation in Attend2Pack allows the solution to vary the length of the container. Being able to vary the length of the container removes some complexity from the problem by never having to choose when to open a new bin (and in more complex cases, which bin type to choose). Attend2Pack also does not enforce 100% support: support is guaranteed at the point of placement of the rear-left-bottom corner of an item, but not over the whole underside area of the item. We were able to force support to be 100% in the Jumanji set up by modifying the way empty maximal spaces are generated after an item is placed. As we wanted to show how CG could improve the RL solution, we believe the comparison with the Jumanji RL agent is sufficient to show this. We would expect similar results had a modified version of Attend2Pack been used instead.
>
> We also didn’t compare our results to “Learning Efficient Online 3D Bin Packing on Packing Configuration Trees” since this is an online bin packing problem and only packs a single bin.
>
> # Answers to your questions:
>
> ## Local optima and feasible solutions:
>
> Indeed, the sub-problem is a 3D knapsack problem, when solved using anything but exact solvers, we cannot guarantee the optimality of the solution (as explained in the introduction of the paper).  However, since we initialize with a feasible (but suboptimal) solution (the sequential RL solution) we can always guarantee a final solution that is feasible. In the worst case scenario, the sequential RL solution is the result we’ll get at the end of the CG algorithm. We see that this is the case for some simpler instances, but not for the more challenging ones. The final paragraph of the abstract mentions lower and upper bounds more clearly to reduce confusion around feasible solutions.

---

> ### Comment · Reviewer_zq6L · 2023-11-23
>
> I have read the authors' responses as well as the comments and responses from other reviewers. Previously, I had concerns that the authors compared too few baselines and used outdated datasets. The authors defended their choice by stating that existing methods and datasets for offline bin packing primarily focus on solving single-container bin packing. However, for offline packing, the boundary between single-container and multi-container problems is not significant. The only difference lies in the additional decision of allocating objects to specific containers in multi-container scenarios, which is easily achievable using heuristic rules or an additional learning-based strategy. Furthermore, packing datasets are not limited to the settings of single or multiple containers, thus the authors' argument lacks persuasiveness. I still believe that it is crucial for the authors to compare their proposed method with end-to-end reinforcement learning algorithms and to conduct tests on updated datasets.
>
> In conclusion, I maintain my previous rating and firmly reject this paper.

---

### Official Review · Reviewer_kDhg · 2023-11-01

**Soundness:** 3 good
**Presentation:** 1 poor
**Contribution:** 2 fair
**Rating:** 3
**Confidence:** 4

**Summary:**

The paper proposes an integration of deep reinforcement learning (RL) in a Dantzig-Wolfe decomposition method to solve the 3D bin packing problem. The idea is to replace the solution of the subproblem in the Dantzig-Wolfe decomposition by an RL agent.

**Strengths:**

The combination of deep reinforcement learning with Dantzig-Wolfe decomposition is novel as far as I know.

The authors obtained some novel lower bounds for some instances of the 3D bin packing problem.

Promising results are obtained on a dataset of 47 instances.

**Weaknesses:**

Although the combination is novel, the novelty in the machine learning part seems to be limited.

It seems that some important information is missing in the presentation such as:
- what are "normal patterns"?
- the architecture of the deep RL agent
- the training methodology of the RL agent
- the sequential RL agent should be explained more explicitly

The formatting in the paper has many issues. For instance:
a2c -> A2C
Bonnet et al. (2023) -> (Bonnet et al., 2023)
Zhang et al. (2021) is limited -> The method proposed by Zhang et al. (2021) is limited
Fang J (2023) -> Fang and Rao (2023)
The names in the citation "Deidson Vitorio Kurpel" should be fixed.
(1-5) and (6-10) should be better indented
Jumanji bin pack -> Jumanji binpack
"Where v_i represents" this sentence should be one sentence with the previous one.
What is a "liquid volume"?

**Questions:**

What architecture did you use for your RL agent?

How did you train it? Which RL algorithm did you use?

Could you include in Table 1 the aggregated results for the other SOTA methods so that it is easier to understand how well the proposed methodology performs?

---

> ### Author Response · Authors · 2023-11-17
> **Thank you for pointing out the strengths you found in our paper and for the very valuable and constructive feedback. We would like to answer your questions in the following reply and note how we have addressed the weaknesses of the initial submission in its revised form.**
>
> Thank you for pointing out the strengths of our paper and for your valuable feedback. We explain in this reply how we have addressed the weaknesses of the original submission in our latest version. We then answer your questions.
>
> # Weaknesses addressed in the latest version of the paper:
>
> ## Novelty:
>
> The final paragraph of the introduction now details all the novel contributions of the paper more clearly. You rightly point out that our main contribution is combining RL with column generation. However we do have novel contributions to the RL part of the work too. A “value” feature was added to the items and used in the reward function. Previous literature optimized for volume utilization and not value packed. New constraints not fully taken into account by previous RL implementations of the knapsack problem were added. RL algorithms have not previously been tested on off-line test cases that require multiple bins to pack all items. This novel contribution shows the need for either increasing the action space or combining the 3D knapsack RL with the column generation algorithm as we have done in this paper.
>
> ## We now provide more detailed explanations of certain points thanks to your feedback:
>
> Section 4.1 gives a brief summary of the deep RL algorithm used and full details of the architecture are given in Appendix C.
>
> We added a brief description of “normal patterns” in appendix E:
> > Normal patterns are a discretization scheme that is used to discretize the three axes of the container that consider all the possible points of placing any item along one axis of the container. They were first used by Junqueira et al. (2012) to solve the problem of 3D knapsack with cargo stability and load-bearing constraints. Kurpel et al. (2020) then extended these patterns to include all possible rotations of items. A full mathematical explanation of these patterns can be found in either reference.
>
> Section 3.3 explains what we mean by sequential RL:
> > ...applying an RL agent (greedy or fine-tuned) on the problem as if it was a knapsack problem until no more items fit into the bin, the remaining items are then taken to form a new knapsack problem. This process is continued until all items are packed.
>
> “Liquid volume” is a term often used as a concise way of saying the sum of the volume of all items in an instance. Your comment has made us realize that this may cause confusion, therefore we have replaced “liquid volume” in section 3.2 by “sum of the volume of all items” .
>
> ## Formatting issues:
> The formatting issues you pointed out should now be fixed. The indentation of equations are now all aligned along the (in)equality. “Jumanji binpack” has been replaced by Jumanji BinPack to be consistent with their github.
>
> # Answers to your questions:
>
> ## What architecture was used for the RL agent, how was it trained and which RL algorithm was used?
>
> We used the same architecture as suggested in the Jumanji github for the BinPack environment, the details are in appendix C. The advantage actor-critic algorithm (A2C) was used to train the deep RL agent; more details on this can now also be found in appendix C.
>
> ## Aggregated results for other SOTA methods in table 1.
>
> We now see that both SOTA methods perform equally in terms of the number of total bins packed over all instances (note, which algorithm performs best on a given test case may vary). We are not yet at the same performance as SOTA solutions. We do obtain results faster than Kurpel et al. (2020). Zhu et al. (2012) performs better and faster on these test cases than our proposed solution (RL+CG) because they rely heavily on fast and efficient heuristics. These would be more difficult to generalize to test cases with complex constraints than an RL agent.

---

### Meta-Review · Area_Chair_Nb3w · 2023-12-10

**Metareview:**

The paper proposes a novel approach combining deep reinforcement learning (RL) with Dantzig-Wolfe decomposition and column generation (CG) to solve the 3D bin packing problem. The reviewers acknowledge the potential of this approach, but raise concerns about the novelty and clarity of the contributions, as well as the experimental evaluation. Specifically, the novelty in the RL part is concerned, important implementation details are missing. The core method section needs more illustrations and motivation. More comparisons to recent methods and datasets are needed. In response, the authors better highlight the novel contributions, including extending the RL environment, adding new constraints and reward function, and being the first to test RL on multi-bin packing instances. They provide more implementation details. However, comparing to more recent end-to-end RL methods and datasets would strengthen the work. Overall, I would recommend the authors address the remaining concerns raised, particularly expanding the experimental comparison, before considering resubmission. The paper in its current form does not seem suitable for acceptance.

**Justification For Why Not Higher Score:**

Limited contribution and missing important baselines.

**Justification For Why Not Lower Score:**

N/A

---

### Decision · Program_Chairs · 2024-01-16

Reject